# First data set of H$_2$O/HDO columns from TROPOMI

Andreas Schneider[1], Tobias Borsdorff[1], Joost aan de Brugh[1], Franziska Aemisegger[2], Dietrich
G. Feist[3,4,5], Rigel Kivi[6], Frank Hase[7], Matthias Schneider[7], and Jochen Landgraf[1]

[1]Earth science group, SRON Netherlands Institute for Space Research, Utrecht, the Netherlands
[2]Atmospheric Dynamics group, Department of Environmental Systems Science, ETH Zürich, Zürich, Switzerland
[3]Ludwig-Maximilians-Universität München, Lehrstuhl für Physik der Atmosphäre, Munich, Germany
[4]Deutsches Zentrum für Luft- und Raumfahrt, Institut für Physik der Atmosphäre, Oberpfaffenhofen, Germany
[5]Max Planck Institute for Biogeochemistry, Jena, Germany
[6]Finnish Meteorological Institute, Sodankylä, Finland
[7]Institute of Meteorology and Climate Research (IMK-ASF), Karlsruhe Institute of Technology, Karlsruhe, Germany

**Correspondence:** Andreas Schneider (a.schneider@sron.nl)

**Abstract.** Global measurements of atmospheric water vapour isotopologues aid to better understand the hydrological cycle and improve global circulation models. This paper presents a new data set of vertical column densities of H$_2$O and HDO retrieved from short-wave infrared (2.3 μm) reflectance measurements by the Tropospheric Monitoring Instrument (TROPOMI) aboard the Sentinel-5 Precursor satellite. TROPOMI features daily global coverage with a spatial resolution of up to $7\,\mathrm{km} \times 7\,\mathrm{km}$.

The retrieval utilises a profile-scaling approach. The forward model neglects scattering, thus strict cloud filtering is necessary. For validation, recent ground-based water vapour isotopologue measurements by the Total Carbon Column Observing Network (TCCON) are employed. A comparison of TCCON $\delta$D with ground-based measurements by the project Multi-platform remote Sensing of Isotopologues for investigating the Cycle of Atmospheric water (MUSICA) for data prior to 2014 (where MUSICA data is available) shows a bias in TCCON $\delta$D estimates. As TCCON HDO is currently not validated, an overall correction of

recent TCCON HDO data is derived based on this finding. The agreement between the corrected TCCON measurements and collocated TROPOMI observations is good with an average bias of $(-0.2 \pm 3) \cdot 10^{21}\,\mathrm{molec\,cm^{-2}}$ $((1.0 \pm 7.4)\,\%)$ in H$_2$O and $(-2 \pm 7) \cdot 10^{17}\,\mathrm{molec\,cm^{-2}}$ $((-1.2 \pm 7.4)\,\%)$ in HDO, which corresponds to a mean bias of $(-14 \pm 18)\,\text{‰}$ in a posteriori $\delta$D. The bias is lower at low and mid latitude stations and higher at high latitude stations. The use of the data set is demonstrated with a case study of a blocking anticyclone in northwestern Europe in July 2018 using single overpass data.

## 1   Introduction

Atmospheric water vapour represents the strongest natural greenhouse gas and transports a large amount of energy through latent heat and thus plays a fundamental role in shaping weather and climate (Kiehl and Trenberth, 1997; Harries, 1997). However, uncertainties in the quantification of these two effects are still large and represent one of the key uncertainties in current climate prediction (Stevens and Bony, 2013). To improve on that requires new observations on a global scale and with

a long-term perspective. To this end, satellite observations from space are considered as the most promising approach (Rast et al., 2014).

Constraints for the hydrological cycle are offered by observations of isotopologues of water vapour. Different equilibrium vapour pressures and diffusion constants of different isotopologues lead to isotopic fractionation whenever a phase change occurs. Isotopic fractionation is associated with a partitioning of the heavy and light isotopologues in the different phases, which

depends on the thermodynamic conditions of the environment. The relative abundance of a heavy isotopologue with respect to the light isotopologue in an air parcel is therefore dependent on the source region's temperature and relative humidity, the source water's isotopic composition as well as the entire transport history of the air parcel, including all evaporation, condensation and mixing events (e. g. Dansgaard, 1964; Craig and Gordon, 1965). This makes measurements of water vapour isotopologues a unique diagnostic of the hydrological cycle (Dansgaard, 1964) and a valuable benchmark for the evaluation and

further development of global and regional circulation models (e. g. Joussaume et al., 1984; Hoffmann et al., 1998; Yoshimura et al., 2008; Risi et al., 2010; Pfahl et al., 2012).

The usual notation to describe the isotopological abundance variations is the relative difference of the ratio of the heavy and the light isotopologue, here HDO and $H_2O$, $R_D = c_{HDO}/c_{H_2O}$, to a standard abundance ratio $R_{D,std}$,

$$\delta D = \frac{R_D - R_{D,std}}{R_{D,std}} \tag{1}$$

(Coplen, 2011). The commonly used standard ratio is Vienna Standard Mean Ocean Water (VSMOW), $R_{D,std} = 3.1152 \times 10^{-4}$.

Measurements of atmospheric water vapour isotopologues are not very common. In situ observations are performed from aircrafts and balloons (e. g. Rinsland et al., 1984; Dyroff et al., 2010, 2015; Herman et al., 2014; Sodemann et al., 2017) and on the ground (e.g. Wen et al., 2010; Aemisegger et al., 2012; Bastrikov et al., 2014) using laser spectrometers or cryogenic trapping techniques. Remote sensing instruments exist on the ground and on space or balloon based platforms. The former

are usually Fourier transform infrared (FTIR) spectrometers. Ground stations are often organised in networks. The largest networks are the Total Carbon Column Observing Network (TCCON, Wunch et al., 2011) and the Network for the Detection of Atmospheric Composition Change (NDACC, De Mazière et al., 2018). The data product of the former includes $H_2O$ and HDO, while for measurements by the latter water vapour isotopologues are retrieved by the project Multi-platform remote Sensing of Isotopologues for investigating the Cycle of Atmospheric water (MUSICA, Schneider et al., 2016). From satellites,

$H_2O$ and HDO were first retrieved by Zakharov et al. (2004) using thermal infrared measurements from the Interferometric Monitor for Greenhouse gases (IMG) sensor aboard the Advanced Earth Observing Satellite (ADEOS). Later, this was followed by the Tropospheric Emission Spectrometer (TES) on the Earth Observing System (EOS) Aura satellite (Worden et al., 2006), the Michelson Interferometer for Passive Atmospheric Sounding (MIPAS) aboard European Space Agency (ESA)'s environmental satellite (ENVISAT) (Steinwagner et al., 2007; Payne et al., 2007), the SCanning Imaging Absorption spec-

troMeter for Atmospheric CHartographY (SCIAMACHY) instrument on ENVISAT (Frankenberg et al., 2009; Scheepmaker et al., 2015; Schneider et al., 2018), the Infrared Atmospheric Sounding Interferometer (IASI) aboard the MetOP satellite (Herbin et al., 2009; Schneider and Hase, 2011; Schneider et al., 2016; Lacour et al., 2012), the Greenhouse Gases Observing Satellite (GOSAT) (Frankenberg et al., 2013; Boesch et al., 2013), and the Atmospheric Infrared Sounder (AIRS) on board the NASA Aqua satellite (Worden et al., 2019). The sensitivity of instruments observing in the thermal infrared (IMG, TES,

MIPAS, IASI, AIRS) is very different from that of instruments measuring in the short-wave infrared like SCIAMACHY and

GOSAT. While the former are mainly sensitive in the stratosphere and free troposphere, the latter have good sensitivity in the lower troposphere including the boundary layer. On 13th October 2017, the Tropospheric Monitoring Instrument (TROPOMI) aboard the Sentinel-5 Precursor (S5P) satellite (Veefkind et al., 2012) was launched. It has a short-wave infrared band in heritage of SCIAMACHY with a spectral range of 2305–2385 nm and a spectral resolution of 0.25 nm, yet with a signal-to-noise ratio much better than SCIAMACHY and an unprecedented spatial resolution of $7\,\mathrm{km} \times 7\,\mathrm{km}$ (in the centre of the swath). This work presents a new data set of $H_2O$ and HDO columns from TROPOMI observations starting at first light of the instrument on 9th November 2017. Section 2 introduces the retrieval method. Section 3 presents a ground-based data set to validate the satellite observations against, and the comparison between both data sets is shown in Section 4. Section 5 provides a first insight into the data set's use for studying synoptic-scale variability in the atmospheric branch of the water cycle. Finally, the summary of the results and the conclusions are given in Section 6.

## 2   Retrieval method

The retrievals are performed with the short-wave infrared retrieval algorithm SICOR, which utilises a profile-scaling approach and is described in detail by Scheepmaker et al. (2016); Landgraf et al. (2016); Borsdorff et al. (2014). Below, the most important features are summarised and the specific setup is given.

Using the spectral window from 2354.0 nm to 2380.5 nm (Scheepmaker et al., 2016), the algorithm fits the total columns of $H_2O$, HDO, $CH_4$ and CO together with a Lambertian surface albedo in the form of an order 1 Legendre polynomial. The isotopologue $H_2{}^{18}O$ is included in the forward model but not fitted. A priori profiles of water vapour are adapted from the European Centre for Medium-Range Weather Forecasts (ECMWF) analysis product. Since the ECMWF data product does not distinguish individual isotopologues, $H_2O$, HDO and $H_2{}^{18}O$ profiles are obtained from the water vapour profile by scaling it with the respective average relative natural abundances. That implicitly corresponds to a prior of $\delta$D of $0\,‰$. A priori profiles of $CH_4$ and CO are taken from TM5 simulations (Krol et al., 2005). Scattering cross-sections are taken from HITRAN 2016 (Gordon et al., 2017). The forward model ignores scattering, so that strict filtering for clear-sky scenes is necessary. To this end, collocated measurements from the Visible Infrared Imaging Radiometer Suite (VIIRS) instrument aboard the Suomi National Polar-orbiting Partnership (S-NPP) satellite, which flies in formation with S5P, are used (Siddans, 2016). The cloud cover threshold is 1 % for both inner field of view and outer field of view. Moreover, soundings with high aerosol load are filtered out by a two-band filter as introduced by Scheepmaker et al. (2016); Hu et al. (2018), which in the present configuration requires that the ratio of retrieved methane in bands with weak and strong absorption (2310–2315 nm and 2363–2373 nm, respectively) is between 0.94 and 1.06. Furthermore, scenes with solar zenith angle greater than $75°$ are discarded because they are prone to errors due to more scattering and diffraction effects which are not covered well by the forward model and due to typically low radiances meaning low signal-to-noise ratios.

An exemplary spectral fit and the resulting residuals (which are defined as measured minus modelled radiances) are shown in Fig. 1. The root mean square (rms) residual (cyan horizontal line in panel (b)) is in the order of the rms uncertainty of the radiance (purple horizontal line).

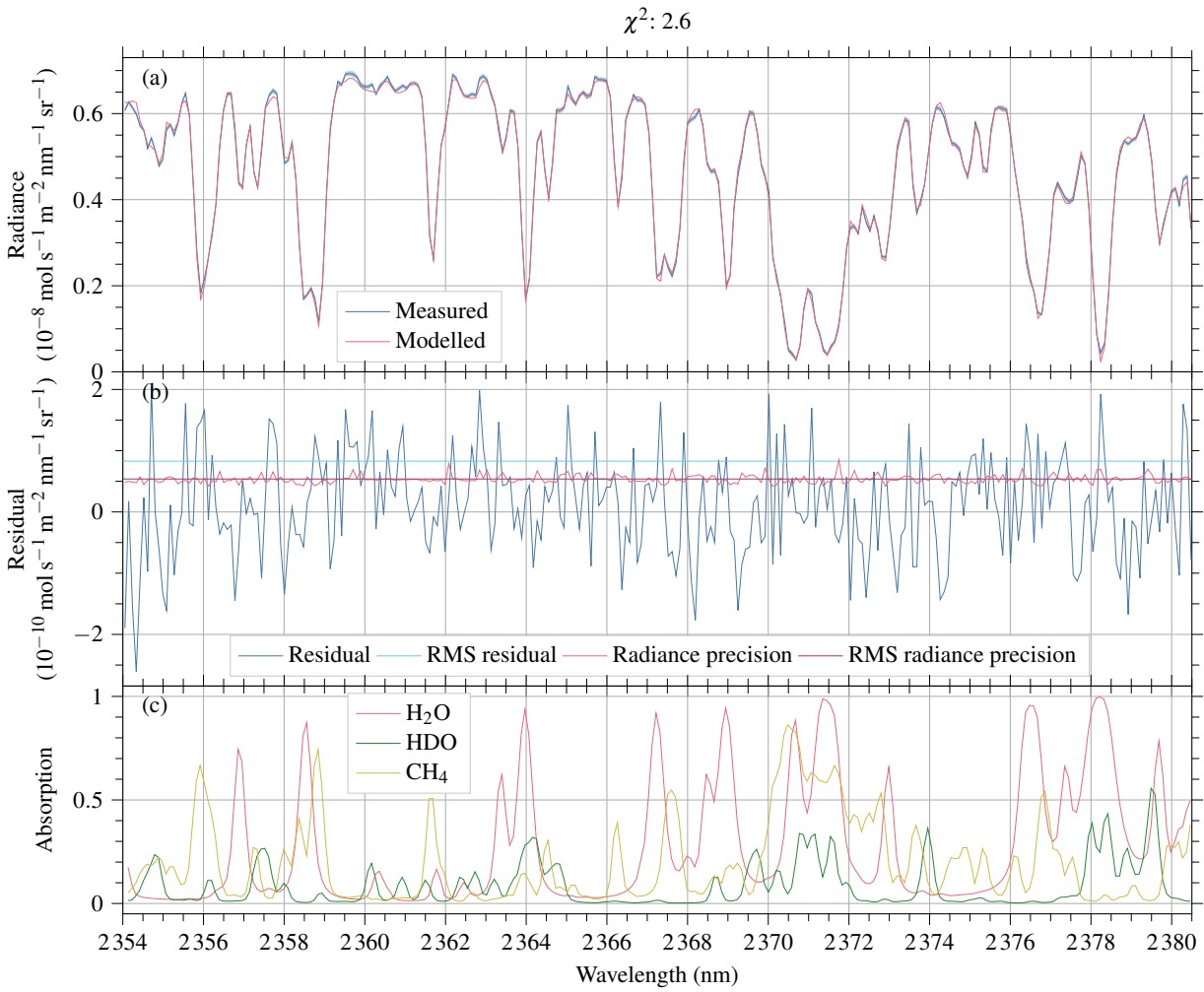

**Figure 1. (a)** Measured radiance (blue) with its precision (light blue band) and spectral fit (red) for ground pixel 149129 in orbit 3969 located near Wollongong, Australia on 20 Jul 2018. **(b)** Corresponding residuals (defined as measured minus modelled radiances, blue) and its root mean square (rms, cyan), precision of the radiance (red) and its rms (purple). **(c)** Simulated absorption by $H_2O$ (red), HDO (green) and $CH_4$ (yellow).

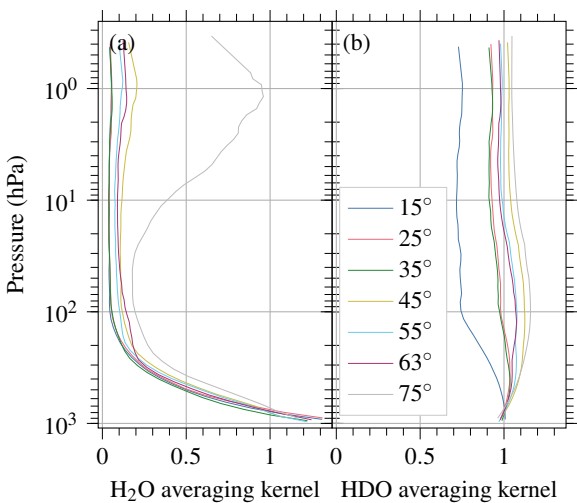

**Figure 2.** Examples of column averaging kernels for **(a)** $H_2O$ and **(b)** HDO for different solar zenith angles in orbit 4924 on 25 Sep 2018.

The sensitivity of a retrieved column to changes in a given altitude region is described by the column averaging kernel
(Rodgers, 2000). The ideal averaging kernel is unity in all altitudes, but in practice the sensitivity changes with height. Figure 2
depicts examples of column averaging kernels for different solar zenith angles. The sensitivity for the two isotopologues are
significantly different. For $H_2O$, the highest sensitivity is in the lowest layer (where typically most water vapour resides) and
decreases with increasing altitude. The sensitivity in the stratosphere is small, however the amount of water vapour in this
altitude region is very small and contributes little to the total column. The sensitivity of HDO does not deviate as much from
unity as the one of $H_2O$. In the lower troposphere it increases slightly with increasing altitude until reaching a maximum
depending on solar zenith angle, above which it decreases. The differences in column averaging kernel are due to the different
absorption strength of the two isotopologues and mean that a posteriori $\delta D$ is sensitive on the profile shapes, particularly of the
main isotopologue $H_2O$ since for that the averaging kernel deviates considerably from unity in higher altitudes.

## 3 Ground-based FTIR data sets

To validate the TROPOMI retrievals, ground-based Fourier transform infrared (FTIR) measurements are used. HDO is a prod-
uct of NDACC-MUSICA (Barthlott et al., 2017) and TCCON (Wunch et al., 2015). NDACC-MUSICA provides two products:
type 1 is the direct retrieval output, and type 2 contains a posteriori processed output which reports the optimal estimation of
$\{H_2O, \delta D\}$ pairs; here the type 2 product is used because it is recommended for isotopologue analyses (Barthlott et al., 2017).
Seven stations are in both networks: Eureka, Ny Ålesund, Bremen, Karlsruhe, Izaña, Wollongong and Lauder. This allows to
compare the TCCON and NDACC-MUSICA (here type 2) data products, which reveals a large difference in $\delta D$ of on average
58 ‰ (which corresponds to a mean relative difference of $-30\,\%$) when collocating with a maximal time difference of one

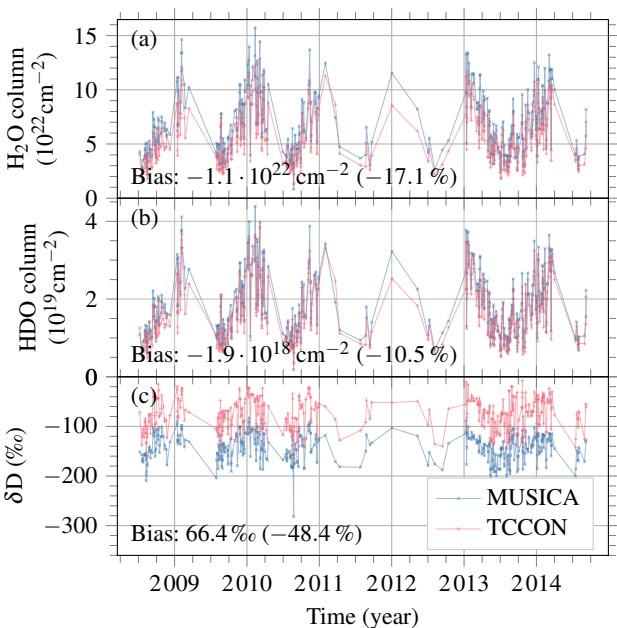

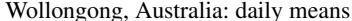

**Figure 3.** Time series of daily averages of $H_2O$ **(a)**, HDO **(b)** and a postiori $\delta D$ **(c)** of the NDACC-MUSICA type 2 (blue crosses) and TCCON (red pluses) data products at Wollongong, Australia. The bias in $\delta D$ without daily averaging is $65.8\,‰$ ($-47.4\,\%$).

hour. An example for Wollongong is plotted in Fig. 3. A comparison between MUSICA and TCCON has also been performed by Weaver (2019). He compares the MUSICA type 1 product with TCCON and finds a bias in $\delta D$ of on average $40\,‰$.

MUSICA is explicitly created for isotopologue studies, and $\delta D$ profiles have been validated against aircraft measurements in
an altitude range of 2–7 km during a dedicated campaign in summer 2013 (Schneider et al., 2015; Dyroff et al., 2015; Schneider et al., 2016). However, data is only available until 2014, so that there is no temporal overlap with TROPOMI which has been launched in October 2017. TCCON $H_2O$ total columns are calibrated with in situ measurements (mainly radiosondes); a so-called aircraft correction factor of 1.0183 is applied to match the reference (Wunch et al., 2015). However, TCCON HDO is currently not verified and so no correction factor is applied to it. Thus it is assumed that TCCON HDO has to be corrected.
In order to correct for the discrepancy, the idea is to scale TCCON HDO to match MUSICA $\delta D$. Scaling HDO with a factor $a$, i. e. $c_{HDO} \mapsto a\,c_{HDO}$, is equivalent to the linear transformation

$$\delta D \mapsto a\,\delta D + a - 1 \tag{2}$$

in $\delta D$. Figure 4a depicts a correlation histogram of TCCON $\delta D$ vs. MUSICA $\delta D$ for the station Wollongong. Here, the relation between MUSICA and TCCON is to a good degree described by a simple scaling of the column. The result of a fit of Eq. (2) to
120 the data is plotted as blue line, giving the scaling factor for the TCCON HDO column. To demonstrate that this approach does not involve intercept issues, a linear fit of slope and intercept (red line) together with the confidence interval computed with

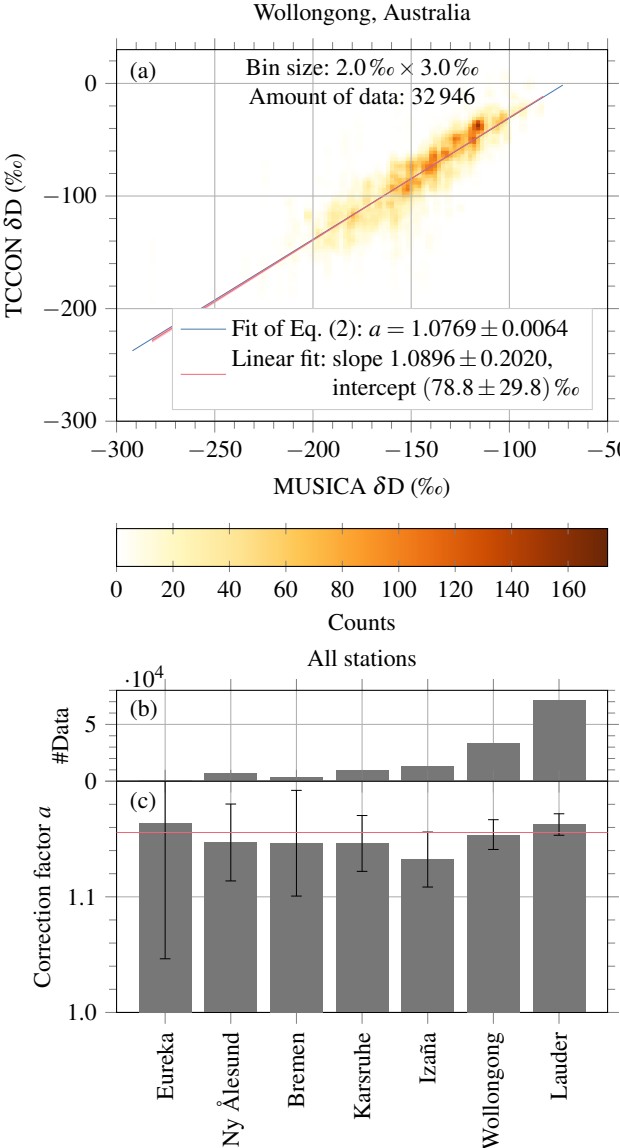

**Figure 4. (a)** Exemplary correlation histogram of TCCON $\delta$D vs. MUSICA $\delta$D for Wollongong, Australia. The blue line shows the result of a fit of Eq. (2), giving the correction factor for TCCON HDO. The red line shows a linear fit with slope and intercept, the red shading its confidence interval computed with the bootstrap method. **(b)** Amount of collocated measurements for all stations in both networks. **(c)** Fit results of correction factors for individual stations. The red line corresponds to the error-weighted average over all stations, $a = 1.0778$.

the bootstrap method (i. e. by 10 000 times fitting a randomly reduced data set, red shading) has also been plotted in the figure. Both slope and offset are similar to the approach with Eq. (2), and the latter lies within the confidence band of the former. The bar chart in Fig. 4c visualises fit results for all stations in both networks. It shows that the correction factor does not change much between stations. The large difference in fit error is mostly due to the large difference in the amount of data (Fig. 4b). Thus it is meaningful to scale HDO at all TCCON stations by the error-weighted average correction factor $a = 1.0778$ in order to correct TCCON's bias in HDO and thus $\delta$D.

**Table 1.** List of TCCON stations used for the validation.

| Station | Latitude | Longitude | Altitude | Data available from/to | Reference |
|---|---|---|---|---|---|
| Eureka | 80.1° N | 86.4° W | 610 m | 24 Jul 2010 – 15 Aug 2019 | Strong et al. (2019) |
| Sodankylä | 67.4° N | 26.6° E | 190 m | 16 May 2009 – 22 Sep 2019 | Kivi et al. (2014) |
| East Trout Lake | 54.4° N | 105.0° W | 500 m | 07 Oct 2016 – 02 Jun 2019 | Wunch et al. (2018) |
| Bialystok | 53.2° N | 23.0° E | 190 m | 01 Mar 2009 – 29 Sep 2018 | Deutscher et al. (2015) |
| Bremen | 53.1° N | 8.9° E | 30 m | 22 Jan 2010 – 27 Sep 2018 | Notholt et al. (2014) |
| Karlsruhe | 49.1° N | 8.4° E | 110 m | 19 Apr 2010 – 02 Jul 2019 | Hase et al. (2015) |
| Paris | 48.8° N | 2.4° E | 60 m | 23 Sep 2014 – 27 Sep 2018 | Té et al. (2014) |
| Orléans | 48.0° N | 2.1° E | 130 m | 29 Aug 2009 – 29 Sep 2018 | Warneke et al. (2019) |
| Park Falls | 45.9° N | 90.3° W | 440 m | 02 Jun 2004 – 03 Jun 2019 | Wennberg et al. (2017) |
| Rikubetsu | 43.5° N | 143.8° E | 380 m | 16 Nov 2013 – 26 Sep 2018 | Morino et al. (2018b) |
| Lamont | 36.6° N | 97.5° W | 320 m | 06 Jul 2008 – 02 Jun 2019 | Wennberg et al. (2016) |
| Tsukuba | 36.0° N | 140.1° E | 30 m | 04 Aug 2011 – 28 Sep 2018 | Morino et al. (2018a) |
| Edwards | 35.0° N | 117.9° W | 700 m | 20 Jul 2013 – 02 Jun 2019 | Iraci et al. (2016) |
| JPL | 34.2° N | 118.2° W | 390 m | 19 May 2011 – 14 May 2018 | Wennberg et al. (2014) |
| Pasadena | 34.1° N | 118.1° W | 240 m | 20 Sep 2012 – 01 Jun 2019 | Wennberg et al. (2015) |
| Saga | 33.2° N | 130.3° E | 10 m | 28 Jul 2011 – 30 Mar 2019 | Kawakami et al. (2014) |
| Wollongong | 34.4° S | 150.9° E | 30 m | 25 Jun 2008 – 29 Sep 2018 | Griffith et al. (2014) |
| Lauder | 45.0° S | 169.7° E | 370 m | 02 Feb 2010 – 03 Apr 2019 | Sherlock et al. (2014); Pollard et al. (2019) |

## 4 Validation of TROPOMI retrievals

For validation, TROPOMI observations are collocated with TCCON measurements with a radius of 30 km, a maximal altitude difference of 500 m, a field of view of 45° in the FTIR viewing direction and a maximal time difference of 2 h. Here, the TCCON HDO data are corrected according to the approach presented in the previous section. Table 1 gives an overview of all stations used. Other stations have too few (less than 5 days) collocated measurements and have thus not been included in the validation study. No altitude correction is applied here. The mentioned collocation criterion for altitude is used to ensure that no bias due to large height difference between station and satellite ground pixel is introduced (cf. Schneider et al., 2018). For each

station, daily averages are computed over all collocated measurements. Figure 5 shows an exemplary time series for Edwards. The collocated observations of $H_2O$ and HDO agree very well, and the agreement in $\delta D$ is also good with more scatter and a small bias. Corresponding correlation plots are depicted in Fig. 6. Panels (a) and (b) confirm the excellent agreement in $H_2O$ and HDO with a Pearson correlation coefficient of 0.99, and a corresponding correlation coefficient of 0.96 for $\delta D$. The average difference between TROPOMI and TCCON defines the bias. In $\delta D$, a small bias is plain in the correlation plot and amounts to $-23\text{\textperthousand}$.

Fig 7 depicts the validation statistics for all TCCON stations. The correlation in $H_2O$ and HDO is high for all stations. In $\delta D$, the correlation is high except for a lower correlation of 0.60 at Lauder and a low correlation of 0.37 at Saga. At these stations the variability in $\delta D$ is small, but $H_2O$ and HDO vary considerably. At Saga the amount of data is very small. Fig. 8 shows the biases. At low and mid latitude ($< 54°$) stations the bias is as low as $(-0.3 \pm 3) \cdot 10^{21}\,\text{molec cm}^{-2}$ (corresponding to a relative bias of $(0.5 \pm 5.8)\,\%$) in $H_2O$ and $(-2 \pm 8) \cdot 10^{17}\,\text{molec cm}^{-2}$ $((-0.6 \pm 6.3)\,\%)$ in HDO, which corresponds to $(-9 \pm 12)\text{\textperthousand}$ $(4.6 \pm 7.2\,\%)$ in a posteriori $\delta D$. At these stations the bias in $\delta D$ ranges between about $-30\text{\textperthousand}$ and $+15\text{\textperthousand}$. At high latitude stations it can be as high as $-45\text{\textperthousand}$ to $-60\text{\textperthousand}$. A reason for these high biases are higher relative biases in $H_2O$ and/or HDO at these relatively dry locations. At high latitudes retrievals are generally challenging due to high solar zenith angles and low albedos which lead to low signal-to-noise ratios. The average bias over all stations is $(-0.2 \pm 3) \cdot 10^{21}\,\text{molec cm}^{-2}$ or $(1.0 \pm 7.4)\,\%$ in $H_2O$, $(-2 \pm 7) \cdot 10^{17}\,\text{molec cm}^{-2}$ or $(-1.2 \pm 7.4)\,\%$ in HDO, and $(-14 \pm 18)\text{\textperthousand}$ or $(5.5 \pm 7.0)\,\%$ in a posteriori $\delta D$. This is good taken into account that $\delta D$ is very sensitive to small errors in $H_2O$ or HDO.

## 5   Demonstration of applications of the data set

An illustration of the TROPOMI retrievals on the global and monthly scale is depicted in Fig. 9 for September 2018. There is no data over the oceans because water is too dark in the short-wave infrared and glint measurements are not taken into account. The data gaps in tropical regions are due to persistent clouds. The data quality in terms of noise is significantly better than for a multi-year average of SCIAMACHY observations, cf. Schneider et al. (2018, Fig. 7). In the spatial distribution shown in Fig. 9 the major isotopic effects formulated by Dansgaard (1964) can be recognised. The general latitudinal gradient due to the temperature-dependence of the fractionation effects and progressive rain out of heavy isotopologues, the so-called latitudinal effect, is clearly visible. The continental effect of depletion due to rain out of the heavy isotopologue is visible on all continents including Australia. The altitude effect, which describes depletion above high ground due to lower temperature and increasing rain out, can be seen, for example, over the Andes and the Himalayas.

To demonstrate the quality and the possibilities of the new data set of water vapour isotopologues from TROPOMI, a case study using single overpass results over Europe on 30th July 2018 is presented in Fig. 10. The summer 2018 was one of the hottest and driest in central and northern Europe (Copernicus Climate Service, 2018; Gubler et al., 2018) with forest fires in Scandinavia, dry fields and low river stages all over the central and northern parts of the continent. The reason for this exceptionally hot and dry summer was the presence of a high pressure system over northern Europe that blocked the otherwise predominant westerly moist flow from the North Atlantic. Synoptic-scale atmospheric blocking situations can lead

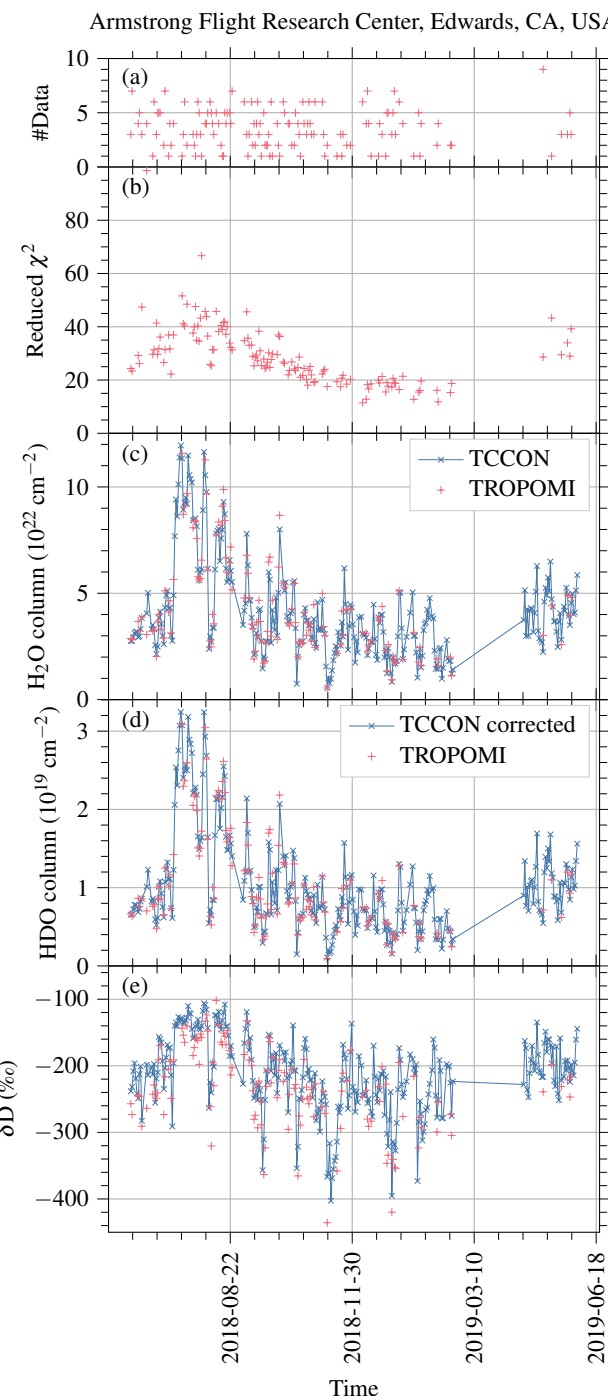

**Figure 5.** Time series of daily averages of corrected TCCON measurements (blue crosses) and collocated TROPOMI observations (red pluses) at Edwards (34.96° N, 700 m a. s. l.). Shown are **(a)** the number of individual observations per day, **(b)** Reduced $\chi^2$, **(c)** $H_2O$ columns, **(d)** HDO columns, and **(e)** a posteriori $\delta D$.

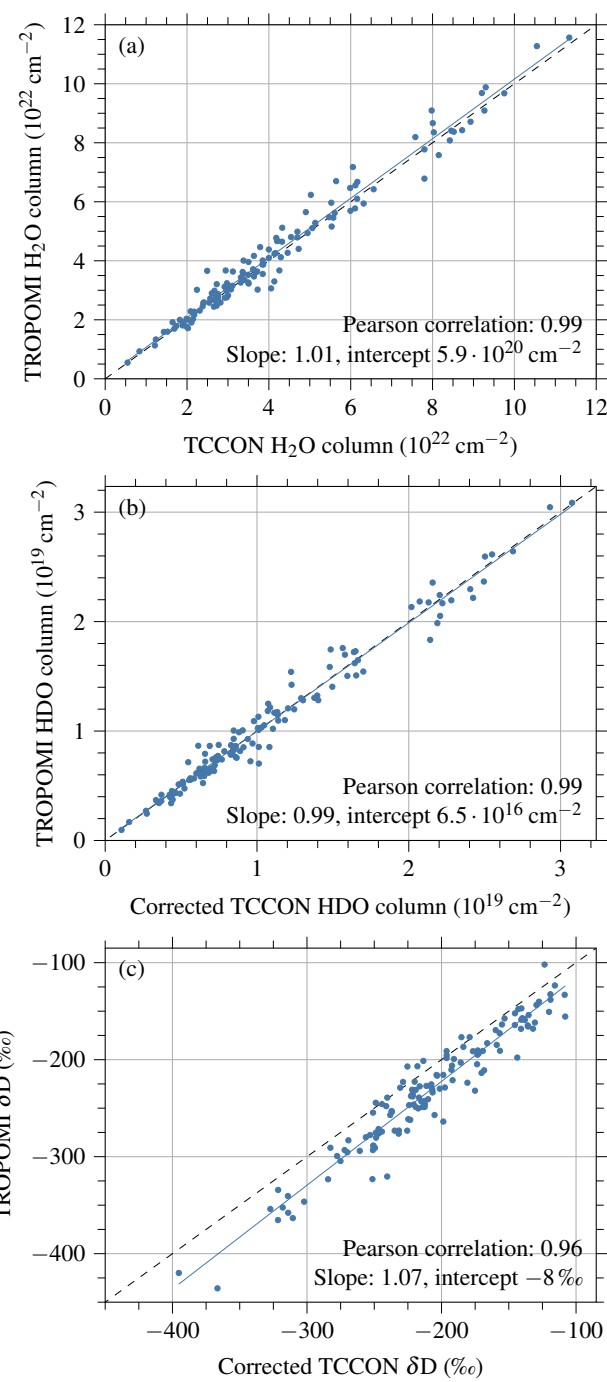

**Figure 6.** Correlation plot of corrected TCCON measurements and collocated TROPOMI observations for Edwards for daily averages of $H_2O$ columns (a), HDO columns (b), and $\delta D$ (c). The dashed lines mark equality, the solid lines give linear fits to the data.

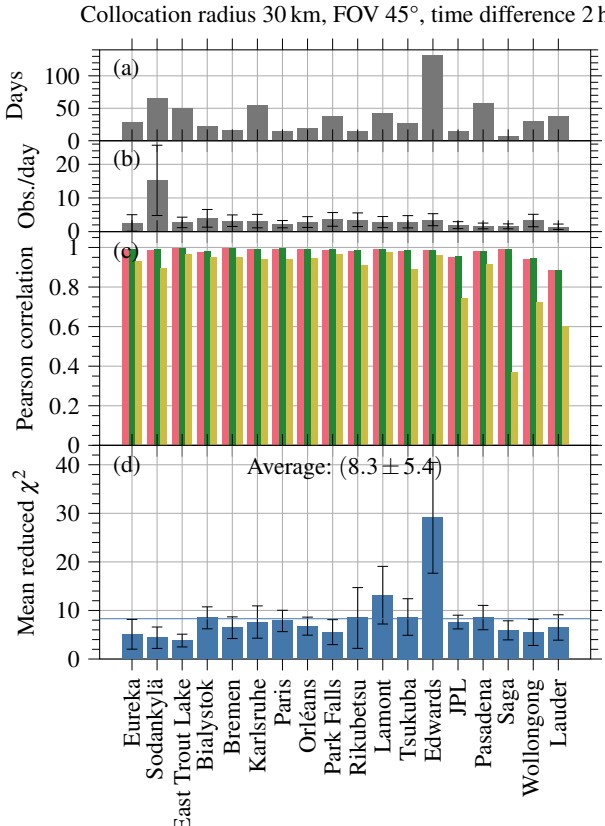

**Figure 7.** Statistics of the validation for all TCCON stations. **(a)** Number of days with collocated measurements. **(b)** Average number of collocated TROPOMI observations per day with its standard deviation. **(c)** Pearson correlation coefficient for $H_2O$ (red), HDO (green) and $\delta D$ (yellow). **(d)** Average reduced $\chi^2$ and its standard error; the blue line visualises the average over all stations.

to hot temperature extremes due to adiabatic warming of the descending air in the core of the anticyclone (Pfahl and Wernli, 2012). The descending vertical motion favours clear sky conditions and thus further contributes to the surface warming through
radiative effects in the centre of the anticyclone (Trigo et al., 2004). In particular, the end of July 2018 was characterised by a stationary blocking anticyclone extending over the entire troposphere over northwestern Russia and Scandinavia. This blocking led to large-scale descent and to a divergent flow near the surface in its core, resulting in clear sky conditions over northwestern Russia and Finland (see Fig. 10c). The isotopic signature of the blocking anticyclone in Fig. 10b reflects this synoptic flow configuration with low $\delta D$ signals of between $-250\,‰$ and $-200\,‰$ in the centre of the anticyclone. The depleted total column
vapour in this region is due to the large-scale subsidence bringing depleted (Fig. 10b) and dry (Fig. 10d) upper tropospheric air towards lower levels. The near-surface divergent wind exports more enriched freshly evaporated moisture that is taken up near the surface towards the edges of the blocking. The anticyclone area is characterised by clear skies (Fig. 10c) with low specific humidity (1–3 g kg$^{-1}$ at 700 hPa, Fig. 10d), low relative humidity (10–30 % at 700 hPa, Fig. 10e) and high potential temperature

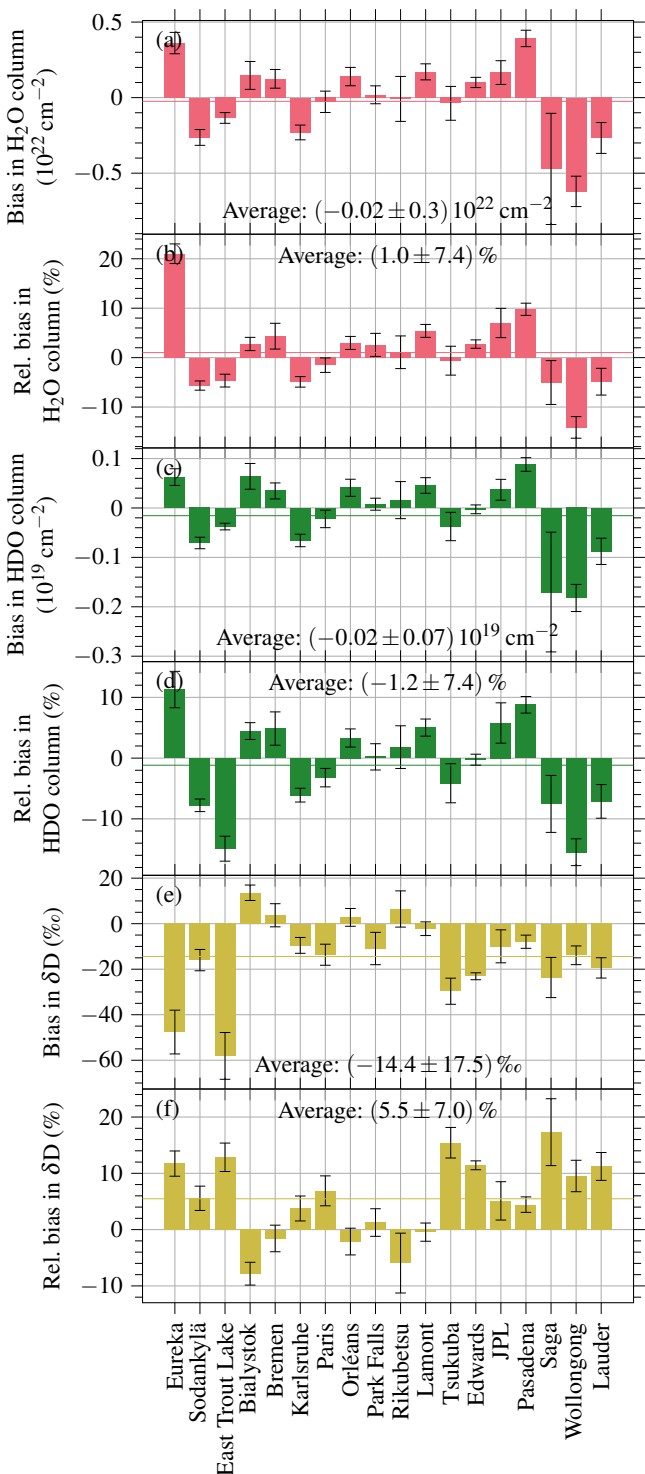

**Figure 8.** Biases for all TCCON stations. **(a)** Bias in $H_2O$ and its standard error. **(b)** Relative bias in $H_2O$ and its standard error. **(c)** Bias in HDO and its standard error. **(d)** Relative bias in HDO and its standard error. **(e)** Bias in a posteriori $\delta D$ and its standard error. **(f)** Relative bias in a posteriori $\delta D$ and its standard error. The horizontal line in all panels visualises the average over all stations.

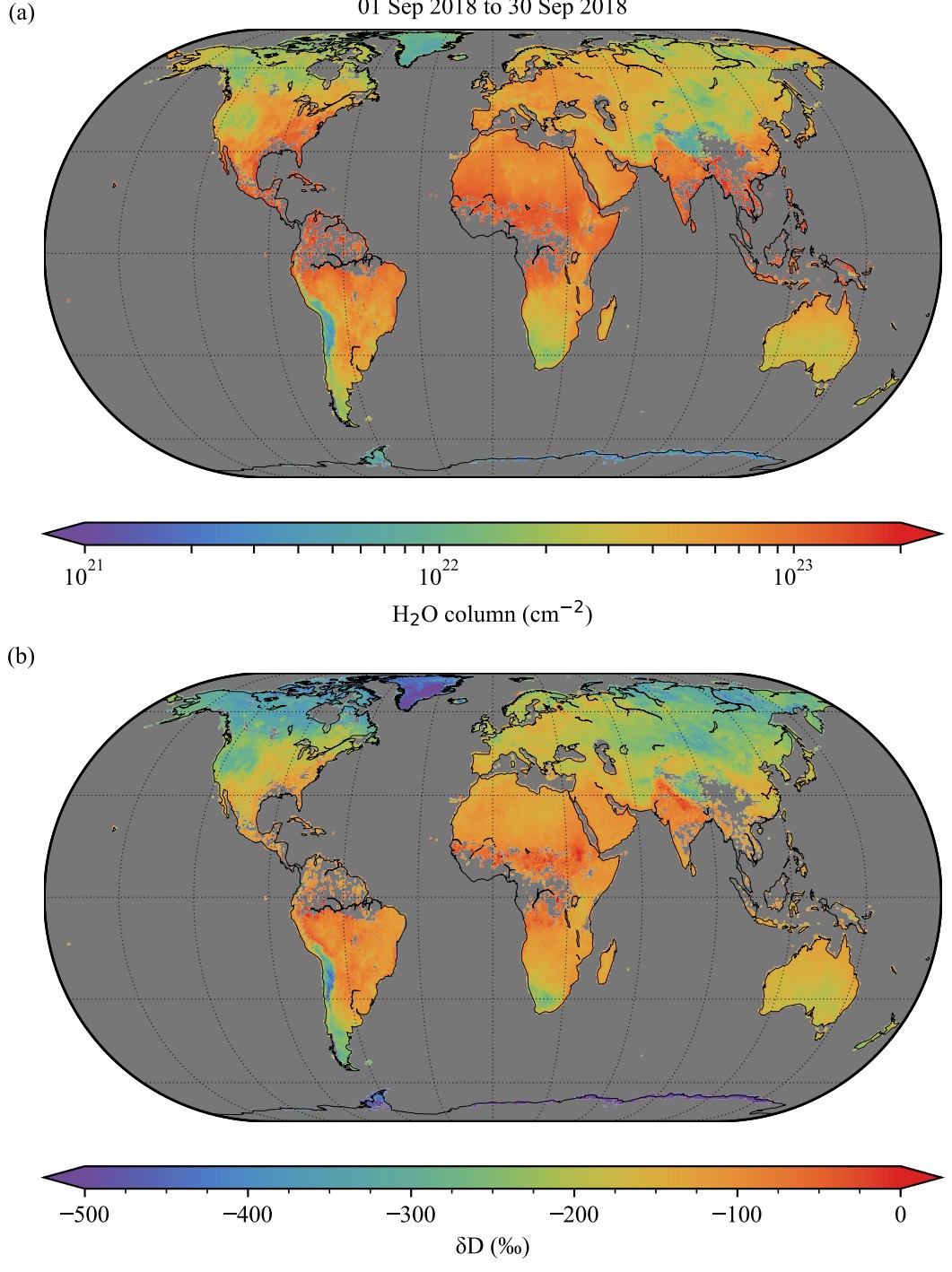

**Figure 9.** Global plots of $H_2O$ **(a)** and $\delta D$ **(b)** averaged over September 2018 on a grid of $0.5° \times 0.5°$. The average of $\delta D$ is weighted with the $H_2O$ column for mass conservation purposes.

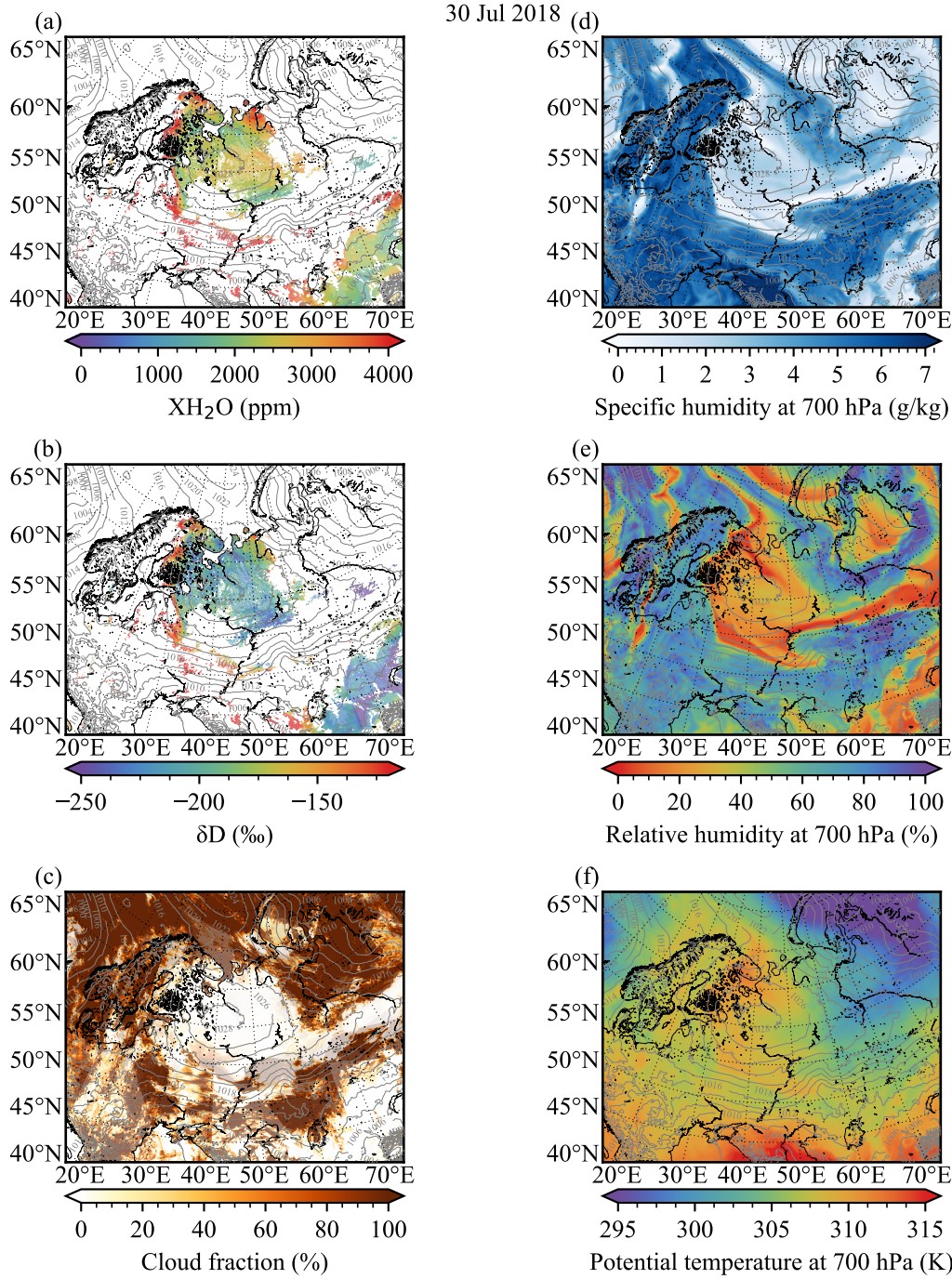

**Figure 10.** TROPOMI single overpass results for H$_2$O column **(a)** and $\delta$D **(b)** over Europe on 30 Jul 2018; VIIRS cloud fraction on the same day **(c)**; specific humidity **(d)**, relative humidity **(e)**, and potential temperature **(f)** at 700 hPa from the ECMWF analysis product over Europe at 12:00 UTC on 30 Jul 2018. The 700 hPa level is chosen for the thermodynamic variables because it reflects the large-scale conditions in the lower troposphere above the continental boundary layer. The overlaying contours in all panels show mean sea-level pressure from ECMWF at 12:00 UTC with a contour line distance of 2 hPa.

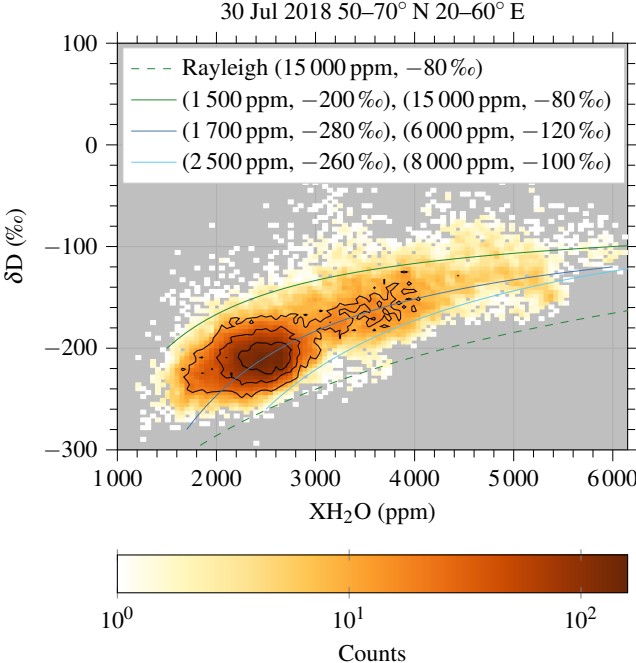

**Figure 11.** Two-dimensional histogram of all TROPOMI observations in the area 50–70° N, 20–60° E on 30 Jul 2018 (colour-coded). The black contours show 25 %, 50 % and 75 % levels of the cumulative density. The dashed green curve represents a Rayleigh fractionation process, the solid green, blue and cyan curves represent idealised mixing processes as specified by the legend.

associated with the dry subsiding (adiabatically warming) air masses (Fig. 10f). The dry low-level outflow encounters moister
and warmer air at the edge of the surface anticyclone, leading to a very strong horizontal gradient of specific and relative
humidity (Fig. 10d,e) in the lower troposphere. As a consequence the warm moist air is forced to rise, localised instabilities
occur and isolated convective cells develop leading to condensation and the formation of a ring of clouds around the blocking
anticyclone. A distinct arc-like feature of enriched total column water vapour at the edge of the anticyclone can be distinguished
slightly displaced from the first clouds in the northwest (Fig. 10b). Turbulent mixing and convection injecting more enriched,
freshly evaporated moisture advected with the large-scale flow from marine environments (Barents Sea, North Sea and Black
Sea) could be the reason for this interesting enriched ring-like water vapour isotopologue pattern. A very depleted cloud free
area south of the Ob river with $\delta$D values below $-250$‰ (Fig. 10b) might be connected to anomalously strong subsidence of
northerly continental air masses.

The large range of measured $\delta$D and $H_2O$ mixing ratios during the northeastern European blocking on 30th July 2018
(Fig. 10a, b) becomes apparent in Fig. 11, where a two-dimensional histogram and the cumulative density of the TROPOMI data
in the region 50–70° N and 20–60° E is shown together with two different types of idealised scenarios of airmass transformation
(coloured lines). These simple idealised scenarios are frequently used in the literature to guide the interpretation of stable
isotope measurements in $H_2O$-$\delta$D diagrams (e. g. Rozanski and Sonntag, 1982; Worden et al., 2007; Noone, 2012). The first

scenario illustrated by the dashed green line in Fig. 11 is that an air parcel with a humidity of 15 000 ppm and $\delta D = -80\,‰$
(typical for the continental boundary layer in this northerly continental region, Bastrikov et al., 2014) experienced moist adiabatic ascent with condensation following a Rayleigh process (dashed green line, Rayleigh, 1902; Dansgaard, 1964). The progressive decrease of $\delta D$ with decreasing water vapour mixing ratio would thus be due to preferential condensation of HDO compared to $H_2O$ and subsequent removal of hydrometeors by precipitation. The dashed green Rayleigh curve in Fig. 11 shows a behaviour that is different from the TROPOMI data points. Given the clear sky conditions and the subsidising movement of the air masses within the blocking anticyclone, the assumptions needed for a Rayleigh distillation process are hardly fulfilled. The second scenario illustrated by the solid green, blue, and cyan lines is that two air parcels with distinct humidity and $\delta D$ are mixed due to turbulent and convective mixing to yield different blends that follow the so-called mixing lines in the $H_2O$-$\delta D$ space. The highest density of observed blocking anticyclone points retrieved by TROPOMI is located in the region spanned by the green and the cyan mixing lines in the $H_2O$-$\delta D$ space. The 25 %, 50 % and 75 % contours of cumulative density of points are aligned with the blue mixing line. This suggests that a two end-member mixing process describes the data much better than an idealised Rayleigh process (dashed green line in Fig. 11). In this particular synoptic situation, this corresponds to the moistening of a subsidising air mass from the mid troposphere.

In future work, the nature and occurrence of these features should be analysed in more detail including a catalogue of different continental blocking events with observations from TROPOMI.

Apart from investigations on the water cycle dynamics associated with continental blockings, many other dynamically interesting contexts exist where TROPOMI could present an important added value for further investigations. These comprise among others the region of the heat low over the Sahara (e. g. Schneider et al., 2015; González et al., 2016; Lacour et al., 2017) or continental regions upstream of cold air surges leading to events of strong ocean evaporation along the warm ocean western boundary currents (Aemisegger and Papritz, 2018; Aemisegger and Sjolte, 2018).

## 6   Summary and conclusions

This work presents a new data set of $H_2O$ and HDO columns retrieved from TROPOMI short-wave infrared observations. Scattering is ignored in the forward model so that a strict cloud filtering is necessary, which is performed with collocated VIIRS measurements. The data quality is such that single overpasses yield meaningful results, which is a huge step forward compared to previous missions like SCIAMACHY.

For validation of the TROPOMI data product, particular attention must be given to the reference data sets. At this stage, there are two data products of ground-based observations of the HDO total column available, provided by the TCCON and NDACC-MUSICA networks. Comparing these two data products for stations in both networks reveals a large bias between the ground-based products of on average 58 ‰ in $\delta D$. NDACC-MUSICA was decidedly developed for water vapour isotopologue studies and is validated in $\delta D$ with aircraft measurements, but data are only available until 2014. TCCON provides recent data with temporal overlap with TROPOMI observations and its $H_2O$ total column data product is validated against in situ

measurements, however its HDO data product is not verified. In order to obtain a suitable validation data set, TCCON HDO columns are scaled by a factor of 1.0778 to match the MUSICA $\delta$D over the common observation time period.

Using a collocation radius of 30 km, a maximal altitude difference of 500 m, a field of view of $45°$ and a maximal time difference of 2 h, a good agreement between corrected TCCON measurements and collocated TROPOMI observations is found. The mean bias is $(-0.2 \pm 3) \cdot 10^{21}$ molec cm$^{-2}$ $((1.0 \pm 7.4)\,\%)$ for H$_2$O, $(-2 \pm 7) \cdot 10^{17}$ molec cm$^{-2}$ $((-1.2 \pm 7.4)\,\%)$ for HDO, and $(-14 \pm 18)\,\permil$ $((5.5 \pm 7.0)\,\%)$ for $\delta$D. At low and mid latitude stations the bias in $\delta$D ranges between about $-30\,\permil$ and $+15\,\permil$, while at high latitude stations it can be as high as $-45\,\permil$ to $-60\,\permil$. Retrievals in high latitudes are challenging due to long light paths and low albedos.

The use of the new data set is demonstrated in a case study of an atmospheric blocking event with a single TROPOMI overpass over northeastern Europe on 30th July 2018. Depleted air masses are found in the core of the anticyclone due to subsidence bringing upper tropospheric air towards lower levels. At the edge of the anticyclone a ring of enriched air is observed. A climatological study on the water vapour isotopic signature of continental summer blocking events could provide promising insights into the atmospheric water cycling associated with such systems that frequently lead to heat waves and hot temperature extremes. This case study shows the quality of the new data set and the added value for isotopologue studies, enabling studies on a day-by-day basis with high spatial resolution over continental regions.

Due to the restrictive filter for clear sky scenes, the data coverage is limited. To improve on this, cloudy-sky retrievals over low clouds will be considered in a future work by using a forward model that accounts for scattering. Moreover, a calibration and validation of the TCCON HDO product is necessary. Additionally, it would be beneficial if recent NDACC-MUSICA data would become available. Finally, an improvement in the consistency between the networks would be very valuable.

*Data availability.* The TROPOMI HDO data set of this study is available for download at ftp://ftp.sron.nl/open-access-data-2/TROPOMI/ tropomi/hdo/9_1/. TCCON data are available from the TCCON Data Archive at https://tccondata.org/. MUSICA data are available from ftp://ftp.cpc.ncep.noaa.gov/ndacc/MUSICA/.

*Author contributions.* Andreas Schneider, Tobias Borsdorff, Joost aan de Brugh and Jochen Landgraf made the TROPOMI HDO retrievals and the analysis. Franziska Aemisegger carried out the case study in Section 5. Dietrich G. Feist aided in the search for the cause of discrepancies between (uncorrected) TCCON HDO and TROPOMI HDO. Rigel Kivi and Frank Hase provided TCCON data. Matthias Schneider provided MUSICA data and TCCON data. All authors discussed the results and commented on the manuscript.

*Competing interests.* The authors declare that they have no conflict of interest.

*Disclaimer.* Plots/data contain modified Copernicus Sentinel data, processed by SRON.

*Acknowledgements.* This work was supported by the ESA Living Planet Fellowship project Water vapour Isotopologues from TROPOMI (WIFT). The TROPOMI data processing was carried out on the Dutch national e-infrastructure with the support of the SURF Cooperative. The project MUSICA has been funded by the European Research Council under the European Community's Seventh Framework Programme (FP7/2007–2013)/ERC grant agreement number 256961. KIT acknowledges BMWi for funding TCCON data analysis and delivery through DLR project 50EE1711A. The TCCON project for the Tsukuba site is supported in part by the GOSAT series project. Nicholas Deutscher, David Griffith, Laura T. Iraci, Isamu Morino, Justus Notholt, Christof Petri, Dave Pollard, Kei Shiomi, Kimberly Strong, Yao Té, Thorsten Warneke, Paul Wennberg and Debra Wunch provided TCCON data.

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
