# Peer review of "First data set of H2O/HDO columns from TROPOMI"

_Atmospheric Measurement Techniques, 2019_

## Short Comment (SC1) · 29 Jul 2019

In case of interest, I'd like to note that a detailed comparison of the MUSICA and TCCON water vapour products has been presented in my PhD thesis, which I defended in November 2018. It can be accessed at: http://hdl.handle.net/1807/95942. This work also includes comparisons between these products and radiosondes. A manuscript on this work is in preparation for publication.

---

## Referee Comment (RC1) · Christian Frankenberg (Referee) · 7 Sep 2019

The manuscript "First data set of $H_2O$/HDO columns from TROPOMI" by Schneider et al provides a first glimpse into the excellent data quality of novel isotopologue measurements from the TROPOMI instrument. As a "first data" paper, it provides a well rounded overview of validation and a few examples of HDO/H2O $\delta$D distributions. Overall, this paper is well suited for AMT and should be published after minor revisions. I also apologize for the late review. Please find some comments (High level first, then detailed aspects) below:

High Level:

- I would really like to see at least one spectral fit in a first data paper but am missing a discussion on fit quality, spectral residuals and the like in this manuscript. Please provide a figure showing typical fits, potentially systematic residuals and locations of HDO and H2O lines (e.g. plot Jacobians and spectral fits + residuals, a few examples are enough but this being a "first data" paper, I consider this a must).

- The averaging kernels for $H_2O$ could be an issue for some of the data analysis, especially if parts of the lower column might be blocked by fractional cloud cover. I understand that the authors strictly filter data to alleviate this problem but I am also wondering whether you can make the averaging kernel more uniform for $H_2O$. There might be a few options. A profile fit could help achieving this, even if the degrees of freedom won't be necessarily high (did you ever try)? You might also try to block out the strongest $H_2O$ lines in the retrieval, which might help (the weaker the lines, the more uniform the averaging kernels). In the long run, this could/should be a focus for further retrieval work as it would allow you to relax filter criteria. However, I realize that this is a bigger endeavor and will require more work in potential future papers.

- In many plots, you always discuss and show "biases", which are additive in nature. However, your analysis uses scaling factors, which are multiplicative. Multiplicative biases are natural, as line strengths in databases can be wrong but it also means that additive biases depend on the amount itself. I would change all bias discussions/plots into relative terms (% bias is basically multiplicative). For fits, provide slope and intercept (e.g. in Figure 5).

- I am missing a Rayleigh plot to be honest, maybe pick a few regions and plot H2O column amount vs delta-D? Would be good just to show the general dependence. Could also be plotted along a typical transect.

Detailed minor comments:

- Abstract: Maybe start with a more general scientific scope sentence in the very beginning

- around line 45: make clear that thermal and SW satellites have very different sensitivities.

- line 57: data "are" (data is plural)

- line 65 "with an order 1 Lambertian albedo", be more specific (I know what you mean). Also, is order 1 enough? Did you try higher orders? If not, why not?

- Line 69: Do I see it correctly that your delta-D prior profile is 0?

- Line 77: Diffraction effects are not really the biggest problem I would say (generally more scattering at higher angles, longer light-paths, etc...)

- Figure 1: Plot with pressure on Y-axis would be more representative.

- Figure 2: Again, bias is misleading here. Express it in %, not absolute terms.How about intercept issues?

- Line 126: molcec

- Line 135: "is plain": What do you mean?

- Figure 6: show relative biases, not absolute

Again, I congratulate the larger TROPOMI team for this excellent satellite mission, providing impressive results at a very early stage.

---

## Referee Comment (RC2) · Anonymous Referee #2 · 14 Sep 2019

The manuscript "First data set of H2O/HDO columns from TROPOMI" by Schneider et al. introduces a new product of HDO/H2O  $\delta$ D global distributions measured by the TROPOMI instrument. The manuscript makes an important contribution to the science community and is very well written. It qualifies to be published as is by AMT.

However, I do feel that it will improve the paper significantly if there are more discussions on the information contents and sensitivities of the measurements. This can be shown in the form of jacobian or averaging kernels for vertical sensitivities. I am also concerned about the global  $\delta D$  latitudinal variations being a factor of temperature distributions (see Figure 7). The cause for this was described as "The general latitudinal gradient due to the temperature-dependence of the fractionation effects and progressive rain out of heavy isotopologues, the so-called latitudinal effect, is plain." In general,

thermal signals are sensitive to the surface thermal contrasts, which reflects measurement sensitivity. This often shows up as latitudinal variations in retrieved products or over higher ground (e.g., the Andes and the Himalayas). I don't know if the authors have studied these effects in the spectral region used. If yes, how would you attribute the latitudinal variations caused by temperature effects of the HDO/H2O physics vs the thermal contrast effects of remote sensing?

Also, since TROPOMI and VIIRS are on different platforms, how were the footprints matched? What was the error estimate in this step?

A very minor point: the Suomi National Polar-orbiting Partnership, should be S-NPP or SNPP, not just NPP.

---

## Author Comment (AC3) · 24 Oct 2019

*The manuscript "First data set of H2O/HDO columns from TROPOMI" by Schneider et al. introduces a new product of HDO/H2O $\delta$D global distributions measured by the TROPOMI instrument. The manuscript makes an important contribution to the science community and is very well written. It qualifies to be published as is by AMT.*

We thank the referee for the positive review. In the following, all individual comments are quoted in italics and our response is given below.

*However, I do feel that it will improve the paper significantly if there are more discussions on the information contents and sensitivities of the measurements. This can be shown in the form of jacobian or averaging kernels for vertical sensitivities.*

We have added a plot of a TROPOMI spectrum with spectral fit and a plot of the corresponding residual. We have also added a panel with $\chi^2$ to all time series and statistics plots. Examples of averaging kernels are plotted in Figure 2 (Figure 1 in the discussion paper).

*I am also concerned about the global $\delta$D latitudinal variations being a factor of temperature distributions (see Figure 7). The cause for this was described as "The general latitudinal gradient due to the temperature-dependence of the fractionation effects and progressive rain out of heavy isotopologues, the so-called latitudinal effect, is plain." In general, thermal signals are sensitive to the surface thermal contrasts, which reflects measurement sensitivity. This often shows up as latitudinal variations in retrieved products or over higher ground (e. g., the Andes and the Himalayas). I don't know if the authors have studied these effects in the spectral region used. If yes, how would you attribute the latitudinal variations caused by temperature effects of the HDO/H2O physics vs the thermal contrast effects of remote sensing?*

Sensed radiation of wavelengths up to 3 μm is predominately reflected solar energy (e. g. Tempfli et al., 2009). Our spectral region at 2.3 μm is in this range. In retrievals in the short-wave infrared thermal radiation is usually neglected, for instance for the SCIAMACHY, GOSAT and TROPOMI data products obtained in this spectral range (e. g. Frankenberg et al., 2009, 2013; Scheepmaker et al., 2015; Schneider et al., 2018; Borsdorff et al., 2018; Hu et al., 2018). Furthermore, the latitudinal and altitude variation of $\delta$D corresponds to our expectations; we don't have indications that it may be wrong.

*Also, since TROPOMI and VIIRS are on different platforms, how were the footprints matched? What was the error estimate in this step?*

We use the official TROPOMI VIIRS data product (Siddans, 2016). S5P and S-NPP fly in formation. The latitude and longitude of the VIIRS pixels are transformed into a Cartesian coordinate system $y$ (across-track), $z$ (along-track), with origin at the centre

of the S5P pixel and scaled such that the corners of the nominal FOV have values of $y = \pm 1$ and $z = \pm 1$. This is the basis for identifying which VIIRS pixels fall into a S5P FOV. An average weighted with a spatial response function is computed. A detailed algorithm description can be found in the ATBD (Siddans, 2016). The data product does not give an error estimation.

*A very minor point: the Suomi National Polar-orbiting Partnership, should be S-NPP or SNPP, not just NPP.*

This has been corrected in the manuscript.

**References**

T. Borsdorff, J. aan de Brugh, H. Hu, O. Hasekamp, R. Sussmann, M. Rettinger, F. Hase, J. Gross, M. Schneider, O. Garcia, W. Stremme, M. Grutter, D. G. Feist, S. G. Arnold, M. De Mazière, M. Kumar Sha, D. F. Pollard, M. Kiel, C. Roehl, P. O. Wennberg, G. C. Toon, and J. Landgraf. Mapping carbon monoxide pollution from space down to city scales with daily global coverage. *Atmos. Meas. Tech.*, 11(10):5507–5518, 2018. doi: 10.5194/amt-11-5507-2018.

C. Frankenberg, K. Yoshimura, T. Warneke, I. Aben, A. Butz, N. Deutscher, D. Griffith, F. Hase, J. Notholt, M. Schneider, H. Schrijver, and T. Röckmann. Dynamic processes governing lower-tropospheric $HDO/H_2O$ ratios as observed from space and ground. *Science*, 325 (5946):1374–1377, 2009. doi: 10.1126/science.1173791.

C. Frankenberg, D. Wunch, G. C. Toon, C. Risi, R.A. Scheepmaker, J.-E. Lee, P. O. Wennberg, and J. Worden. Water vapor isotopologue retrievals from high-resolution GOSAT shortwave infrared spectra. *Atmos. Meas. Tech.*, 6(2):263–274, 2013. doi: 10.5194/amt-6-263-2013.

H. Hu, J. Landgraf, R. Detmers, T. Borsdorff, J. aan de Brugh, I. Aben, A. Butz, and O. Hasekamp. Toward global mapping of methane with TROPOMI: First results and intersatellite comparison to GOSAT. *Geophys. Res. Lett.*, 45(8):3682–3689, 2018. doi: 10.1002/2018GL077259.

R. A. Scheepmaker, C. Frankenberg, N. M. Deutscher, M. Schneider, S. Barthlott, T. Blumenstock, O. E. Garcia, F. Hase, N. Jones, E. Mahieu, J. Notholt, V. Velazco, J. Landgraf, and I. Aben. Validation of SCIAMACHY $HDO/H_2O$ measurements using the TCCON and NDACC-MUSICA networks. *Atmos. Meas. Tech.*, 8(4):1799–1818, 2015. doi: 10.5194/amt-8-1799-2015.

A. Schneider, T. Borsdorff, J. aan de Brugh, H. Hu, and J. Landgraf. A full-mission data set of $H_2O$ and HDO columns from SCIAMACHY 2.3 µm reflectance measurements. *Atmos. Meas. Tech.*, 11(6):3339–3350, 2018. doi: 10.5194/amt-11-3339-2018.

R. Siddans. S5P-NPP cloud processor ATBD. Technical report, Rutherford Appleton Laboratory, 2016. URL http://www.tropomi.eu/sites/default/files/files/S5P-NPPC-RAL-ATBD-0001_NPP-Clouds_v1p0p0_20160212.pdf.

K. Tempfli, N. Kerle, G. C. Huurneman, and L. L. F. Janssen, editors. *Principles of Remote Sensing*. The International Institute for Geo-Information Science and Earth Observation, 2009. ISBN 978-90-6164-270-1. URL https://webapps.itc.utwente.nl/librarywww/papers_2009/general/principlesremotesensing.pdf.

---

## Author Response (AR1)

**Author response to review by Christian Frankenberg on "First data set of H2O/HDO columns from TROPOMI" by Andreas Schneider et al.**

The manuscript "First data set of  $H_2O/HDO$  columns from TROPOMI" by Schneider et al provides a first glimpse into the excellent data quality of novel isotopologue measurements from the TROPOMI instrument. As a "first data" paper, it provides a well rounded overview of validation and a few examples of HDO/H2O  $\delta D$  distributions. Overall, this paper is well suited for AMT and should be published after minor revisions. I also apologize for the late review. Please find some comments (High level first, then detailed aspects) below:

The authors would like to thank for the positive and constructive review. In the following, all individual comments are quoted in italics and our response is given below.

**High Level comments**

• I would really like to see at least one spectral fit in a first data paper but am missing a discussion on fit quality, spectral residuals and the like in this manuscript. Please provide a figure showing typical fits, potentially systematic residuals and locations of HDO and H2O lines (e. g. plot Jacobians and spectral fits + residuals, a few examples are enough but this being a "first data" paper, I consider this a must).

Plots of a TROPOMI radiance measurement with spectral fit, the corresponding residual, and absorption of H2O, HDO and CH4 have been added. Moreover,  $\chi^2$  was added to all time series and statistics plots.

In this context, the time series has been extended with new TCCON measurements that have newly become available, which slightly changes the validation results.

• The averaging kernels for H2O could be an issue for some of the data analysis, especially if parts of the lower column might be blocked by fractional cloud cover. I understand that the authors strictly filter data to alleviate this problem but I am also wondering whether you can make the averaging kernel more uniform for H2O. There might be a few options. A profile fit could help achieving this, even if the degrees of freedom won't be necessarily high (did you ever try)? You might also try to block out the strongest H2O lines in the retrieval, which might help (the weaker the lines, the more uniform the averaging kernels). In the long run, this could/should be a focus for further retrieval work as it would allow you to relax filter criteria. However, I realize that this is a bigger endeavor and will require more work in potential future papers.

Setting up a profile retrieval would need a different algorithm and is out of the scope of this paper. Moreover, due to the spectral resolution and the signal-to-noise ratio of the measurement we believe that the information content is not more than one degree of freedom for HDO.

We tried a few different spectral windows, for instance with weaker  $H_2O$  lines, but the averaging kernel of  $H_2O$  changed only little. Furthermore, in the spectral range of the SWIR channel without very strong  $H_2O$  lines there are only very weak HDO lines so that no useful retrieval of HDO is possible there.

• In many plots, you always discuss and show "biases", which are additive in nature. However, your analysis uses scaling factors, which are multiplicative. Multiplicative biases are natural, as line strengths in databases can be wrong but it also means that additive biases depend on the amount itself. I would change all bias discussions/plots into relative terms (% bias is basically multiplicative). For fits, provide slope and intercept (e.g. in Figure 5).

In the revised version, relative biases are shown additionally to the absolute biases.

We have added a fit to the correlation plots in Figure 5 (now Figure 6) and give the respective slope and intercept.

• I am missing a Rayleigh plot to be honest, maybe pick a few regions and plot  $H_2O$  column amount vs delta-D? Would be good just to show the general dependence. Could also be plotted along a typical transect.

We have added a Rayleigh plot for the case study of the blocking anticyclone in Section 5 in form of a histogram with contours for the cumulative density together with a Rayleigh fractionation line and three mixing lines, as well as a discussion of it in the text.

**Detailed minor comments**

• Abstract: Maybe start with a more general scientific scope sentence in the very beginning

We have added the following sentence:

Global measurements of atmospheric water vapour isotopologues aid to better understand the hydrological cycle and improve global circulation models.

• around line 45: make clear that thermal and SW satellites have very different sensitivities.

We have added the following two sentences:

The sensitivity of instruments observing in the thermal infrared (IMG, TES, MIPAS, IASI, AIRS) is very different from that of instruments measuring in the short-wave infrared like SCIAMACHY and GOSAT. While the former are mainly sensitive in the stratosphere and free troposphere, the latter have good sensitivity in the lower troposphere including the boundary layer.

• line 57: data "are" (data is plural)

Not sure what you are referring to. In the sentence in line 57, "the comparison between both data sets is performed in Section 4", the "is" refers to "comparison".

• line 65 "with an order 1 Lambertian albedo", be more specific (I know what you mean). Also, is order 1 enough? Did you try higher orders? If not, why not?

We have rephrased that as "together with a Lambertian surface albedo in the form of an order 1 Legendre polynomial".

We have tried retrievals with different orders. With order 0 the residual is tilted. For orders higher than 1, the difference to order 1 is not so large, and it is beneficial not to waste too many degrees of freedom here.

• Line 69: Do I see it correctly that your delta-D prior profile is 0?

Although  $\delta D$  is not explicitly retrieved, the prior profiles of H2O and HDO result in an implicit prior of  $\delta D$  of 0 %. In the manuscript we have added the following sentence:

That implicitly corresponds to a prior of  $\delta D$  of 0%.

• Line 77: Diffraction effects are not really the biggest problem I would say (generally more scattering at higher angles, longer light-paths, etc...)

We have rephrased this sentence as follows:

Furthermore, scenes with solar zenith angle greater than 75° are discarded because they are prone to errors due to more scattering and diffraction effects which are not covered well by the forward model and due to typically low radiances meaning low signal-to-noise ratio.

• Figure 1: Plot with pressure on Y-axis would be more representative.

The averaging kernel is now plotted in pressure coordinates with a logarithmic pressure axis.

• Figure 2: Again, bias is misleading here. Express it in %, not absolute terms. How about intercept issues?

We have added the relative bias. Not sure what exactly you mean with "intercept issues". We have added a linear fit with slope and intercept in Figure 3a (now Figure 4a) and the confidence interval computed with the bootstrap method. The original fit is included in the confidence interval, thus there are no significant intercept issues. The manuscript has been updated accordingly.

• Line 126: molcec

This typo has been corrected.

• Line 135: "is plain": What do you mean?

We have rephrased this to "is clearly visible".

• Figure 6: show relative biases, not absolute

We additionally show relative biases in the revised version. To this end, we have split the figure in two, one for the statistics and one for the biases.

**Author response to review by Anonymous Referee #2 on "First data set of H2O/HDO columns from TROPOMI" by Andreas Schneider et al.**

The manuscript "First data set of H2O/HDO columns from TROPOMI" by Schneider et al. introduces a new product of HDO/H2O  $\delta D$  global distributions measured by the TROPOMI instrument. The manuscript makes an important contribution to the science community and is very well written. It qualifies to be published as is by AMT.

We thank the referee for the positive review. In the following, all individual comments are quoted in italics and our response is given below.

However, I do feel that it will improve the paper significantly if there are more discussions on the information contents and sensitivities of the measurements. This can be shown in the form of jacobian or averaging kernels for vertical sensitivities.

We have added a plot of a TROPOMI spectrum with spectral fit and a plot of the corresponding residual. We have also added a panel with  $\chi^2$  to all time series and statistics plots. Examples of averaging kernels are plotted in Figure 2 (Figure 1 in the discussion paper).

I am also concerned about the global  $\delta D$  latitudinal variations being a factor of temperature distributions (see Figure 7). The cause for this was described as "The general latitudinal gradient due to the temperaturedependence of the fractionation effects and progressive rain out of heavy isotopologues, the so-called latitudinal effect, is plain." In general, thermal signals are sensitive to the surface thermal contrasts, which reflects measurement sensitivity. This often shows up as latitudinal variations in retrieved products or over higher ground (e. g., the Andes and the Himalayas). I don't know if the authors have studied these effects in the spectral region used. If yes, how would you attribute the latitudinal variations caused by temperature effects of the HDO/H2O physics vs the thermal contrast effects of remote sensing?

Sensed radiation of wavelengths up to 3  $\mu$ m is predominately reflected solar energy [e. g. Tempfli et al., 2009]. Our spectral region at 2.3  $\mu$ m is in this range. In retrievals in the short-wave infrared thermal radiation is usually neglected, for instance for the SCIAMACHY, GOSAT and TROPOMI data products obtained in this spectral range [e. g. Frankenberg et al., 2009, 2013, Scheepmaker et al., 2015, Schneider et al., 2018, Borsdorff et al., 2018, Hu et al., 2018]. Furthermore, the latitudinal and altitude variation of  $\delta D$  corresponds to our expectations; we don't have indications that it may be wrong.

Also, since TROPOMI and VIIRS are on different platforms, how were the footprints matched? What was the error estimate in this step?

We use the official TROPOMI VIIRS data product [Siddans, 2016]. S5P and S-NPP fly in formation. The latitude and longitude of the VIIRS pixels are transformed into a Cartesian coordinate system y (across-track), z (along-track), with origin at the centre of the S5P pixel and scaled such that the corners of the nominal FOV have values of  $y = \pm 1$  and  $z = \pm 1$ . This is the basis for identifying which VIIRS pixels fall into a S5P FOV. An average weighted with a spatial response function is computed. A detailed algorithm description can be found in the ATBD [Siddans, 2016]. The data product does not give an error estimation.

A very minor point: the Suomi National Polar-orbiting Partnership, should be S-NPP or SNPP, not just NPP.

This has been corrected in the manuscript.

**First data set of H2O/HDO columns from TROPOMI**

Andreas Schneider1, Tobias Borsdorff1, Joost aan de Brugh1, Franziska Aemisegger2, Dietrich G. Feist3,4,5, Rigel Kivi6, Frank Hase7, Matthias Schneider7, and Jochen Landgraf1

1Earth science group, SRON Netherlands Institute for Space Research, Utrecht, the Netherlands

2Atmospheric Dynamics group, Department of Environmental Systems Science, ETH Zürich, Zürich, Switzerland

3Ludwig-Maximilians-Universität München, Lehrstuhl für Physik der Atmosphäre, Munich, Germany

4Deutsches Zentrum für Luft- und Raumfahrt, Institut für Physik der Atmosphäre, Oberpfaffenhofen, Germany

5Max Planck Institute for Biogeochemistry, Jena, Germany

6Finnish Meteorological Institute, Sodankylä, Finland

7Institute of Meteorology and Climate Research (IMK-ASF), Karlsruhe Institute of Technology, Karlsruhe, Germany **Correspondence:** Andreas Schneider (a.schneider@sron.nl)

**Abstract.** Global measurements of atmospheric water vapour isotopologues aid to better understand the hydrological cycle and improve global circulation models. This paper presents a new data set of vertical column densities of the water vapour isotopologues-H2O and HDO retrieved from short-wave infrared (2.3 µm) reflectance measurements by the Tropospheric Monitoring Instrument (TROPOMI) aboard the Sentinel-5 Precursor satellite. TROPOMI features daily global coverage with a

- 5 spatial resolution of up to  $7 \text{ km} \times 7 \text{ km}$ . The retrieval utilises a profile-scaling approach. The forward model neglects scattering, thus strict cloud filtering is necessary. For validation, recent ground-based water vapour isotopologue measurements by the Total Carbon Column Observing Network (TCCON) are employed. A comparison of TCCON  $\delta D$  with ground-based measurements by the project Multi-platform remote Sensing of Isotopologues for investigating the Cycle of Atmospheric water (MUSICA) for data prior to 2014 (where MUSICA data is available) shows a bias in TCCON  $\delta D$  estimates. As TCCON
- 10 HDO is currently not validated, an overall correction of recent TCCON HDO data is derived based on this finding. The agreement between the corrected TCCON measurements and collocated TROPOMI observations is good with an average bias of  $(0.02\pm2) \cdot 10^{21}(-0.2\pm3) \cdot 10^{21}$  molec cm-2  $((1.0\pm7.4)\%)$  in H2O and  $(-0.3\pm7) \cdot 10^{17}(-2\pm7) \cdot 10^{17}$  molec cm-2  $((-1.2\pm7.4)\%)$  
[revised manuscript text omitted]